# Molecular imaging of glycan chains couples cell-wall polysaccharide architecture to bacterial cell morphology

Robert D. Turner[1], Stéphane Mesnage[1], Jamie K. Hobbs[1] & Simon J. Foster[1]

Biopolymer composite cell walls maintain cell shape and resist forces in plants, fungi and bacteria. Peptidoglycan, a crucial antibiotic target and immunomodulator, performs this role in bacteria. The textbook structural model of peptidoglycan is a highly ordered, crystalline material. Here we use atomic force microscopy (AFM) to image individual glycan chains in peptidoglycan from *Escherichia coli* in unprecedented detail. We quantify and map the extent to which chains are oriented in a similar direction (orientational order), showing it is much less ordered than previously depicted. Combining AFM with size exclusion chromatography, we reveal glycan chains up to 200 nm long. We show that altered cell shape is associated with substantial changes in peptidoglycan biophysical properties. Glycans from *E. coli* in its normal rod shape are long and circumferentially oriented, but when a spheroid shape is induced (chemically or genetically) glycans become short and disordered.

[1] Krebs Institute, University of Sheffield, Sheffield S10 2TN, UK. Correspondence and requests for materials should be addressed to J.K.H. (email: jamie.hobbs@sheffield.ac.uk) or to S.J.F. (email: s.foster@sheffield.ac.uk)

Peptidoglycan consists of periodic glycan copolymers (N-acetyl glucosamine and N-acetyl muramic acid, Supplementary Fig. 1b) covalently cross-linked by bonds between peptides (often tetra- or penta-peptides, e.g. L-Ala, D-Glu, meso-DAP, D-Ala, D-Ala, Supplementary Fig. 1c). Polymerisation of sugars is catalysed by enzymes known as penicillin-binding proteins (PBPs) and "SEDS" proteins[1]. Peptide cross-links are catalysed by PBPs. Other families of enzymes (hydrolases) break bonds allowing cell shape to be dynamic[2]. The core polymer structural motif is broadly conserved across the bacteria; however, great variation in amino acid composition, degree of cross-linking, glycan chain length and chemical modifications (e.g. acetylation) has been revealed via extensive biochemical studies[3]. Information on how the glycan chains are arranged relative to each other is much more limited. AFM provides high signal-to-noise imaging of individual biomolecules at the nanoscale[4] but despite considerable effort, chains have not previously been visualised in polymer networks[5–10].

The E. coli cell-wall peptidoglycan is both a single macromolecule (sacculus) and a minimal structural material—essentially a two-dimensional polymer network thought to account for most of its mechanical properties[3]. This makes it an excellent model material to study. The qualitative structural model adopted by textbooks and used as the basis for simulations of the polymer envelope[11,12] is that glycan chains run circumferentially around the cylindrical part of the cell and in the same direction at the poles. The measured directional anisotropy in the "Elastic Modulus" of the material is explained as a result of the circumferentially oriented glycans being stiffer than the longitudinally oriented peptide cross-links[9]. As the field moves from conceptual descriptions of how the peptidoglycan polymer envelope fulfils its "physical" roles of maintaining shape and resisting force to sophisticated computational simulations[13,14], allowing detailed predictions of polymer dynamics and interactions with other constituents of the cell, there is a necessity for experimental data on which to build an increasingly accurate physical description of the material.

In this study, we made use of AFM to visualise entire and fragments of peptidoglycan sacculi from E. coli. We directly observed glycan chains in these, and developed a method to quantify the extent to which they are oriented in the same direction. This was used to show that glycan chain orientation is similar in the poles and the cylindrical part of E. coli. We observed very long chains in our images, and confirmed this finding by isolating and purifying chains from peptidoglycan and measuring them by AFM. Finally, we applied our methods to peptidoglycan from spheroid E. coli (produced by chemical and genetic means) showing that these have shorter glycan chains and much reduced chain orientational order compared to the wild-type rods.

## Results

### Direct visualisation of glycans in the peptidoglycan polymer.

Extensive AFM studies of isolated peptidoglycan[5–9] prior to this had failed to resolve glycan chains, a critical step in enabling understanding of how the polymer network performs its function. We therefore developed new approaches to sample preparation and imaging ultimately allowing glycans to be seen.

We extracted peptidoglycan polymer envelopes (sacculi) from E. coli (MG1655)[7], prior to immobilisation on a mica support coated with poly-L-ornithine, without drying at any point. We used standard biochemical purification methods, established over decades. These may appear "harsh", but do not degrade peptidoglycan, the structure of which is mainly determined by covalent bonds between sugars and peptides.

We made both intact sacculi and sacculus fragments, generated by sonication. E. coli sacculus fragments have previously been imaged by TEM[15] and AFM[7]. Sacculi break apart in a predictable way, with cracks forming around the circumferential axis of E. coli cells[7,15]. We found that intact sacculi were less stable than single leaflet fragments under these conditions and could not be imaged (Supplementary Fig. 2a, b). We tested several imaging buffers including 10 mM MES (pH 6, Supplementary Fig. 2c, d) and 10 mM CAPS (pH 10, Supplementary Fig. 2e, f). The addition of salt (KCl) to the buffer caused the peptidoglycan to detach from the surface.

Optimal imaging was obtained with 10 mM Tris (pH 8) (Fig. 1a–d; Supplementary Fig. 1). Firm attachment of the polymer to the substrate was critical for successful imaging[16]. The regions that appear as white patches in 2D representations of the data (e.g. Supplementary Fig. 1e) or as yellow–green–white raised areas in 3D representations (Fig. 1b) are relatively higher than their surroundings and may indicate a small number of chains that were not fully resolved.

Polymer envelope imaging revealed some features with a spacing of 2.7 nm (s.d. 0.5 nm, $n = 19$) consistent with historically predicted values for that between glycan chains (1–4 nm[17], Supplementary Fig. 1c). We measured well resolved chains to be 1.4 nm wide at full width half maximum (s.d. 0.4 nm, $n = 38$). These features had lengths in excess of 10 nm, much too long to be peptide cross-linkers. Densities ≈4 nm in diameter previously observed by electron cryo-tomography (ECT)[18] had been attributed to glycan chains. It is possible, however, that these were multiple parallel chains.

The extent to which Gram-negative peptidoglycan can be multi-layered or of variable thickness is debated in the literature. ECT data suggest a single layer[18], contradicting an earlier neutron scattering study[19], which reported 75–80% of the surface was single-layered. Very recent work invokes variable peptidoglycan thickness to explain response to externally applied stresses[20]. Our data show that although the polymer is generally a single glycan chain thick, some chains overlap each other (Fig. 1d; Supplementary Fig. 1h), indicating that variation in thickness/areal density is possible. Overlapping strands will pertain to the mode of synthesis that we have previously hypothesised whereby areal growth requires both synthesis of material and subsequent hydrolysis to allow wall expansion[21]. Extraordinarily, glycan chains apparently longer than the maximum literature value (60 nm)[22] were seen (Fig. 1c; Supplementary Fig. 1g, h), although we could not locate chain ends for our own measurements within the polymer envelope fragments—this was likely due to tip-induced or thermal motion of the strand ends.

### Glycan orientational order is quantifiable.

All previous studies of peptidoglycan have been restricted to qualitative descriptions of polymer organisation. We investigated several approaches to quantification of the network, before developing a bespoke analytical method achieving statistically meaningful comparisons of peptidoglycan architecture. This would not have been possible without our new chain imaging capability.

We did not observe a regular, crystalline arrangement of features in our images. This was confirmed by Fourier transforms which revealed no distinct spots which could be assigned to a regular, long-range, periodic spacing of the glycan chains (Supplementary Fig. 1i). Despite the absence of peaks in our Fourier transforms, the shape of the somewhat elliptical central spot (Supplementary Fig. 1i) suggested some orientational order, but the nature of contrast in AFM is not particularly compatible with a simple Fourier transform approach. We therefore developed a means of parameterising orientational order by

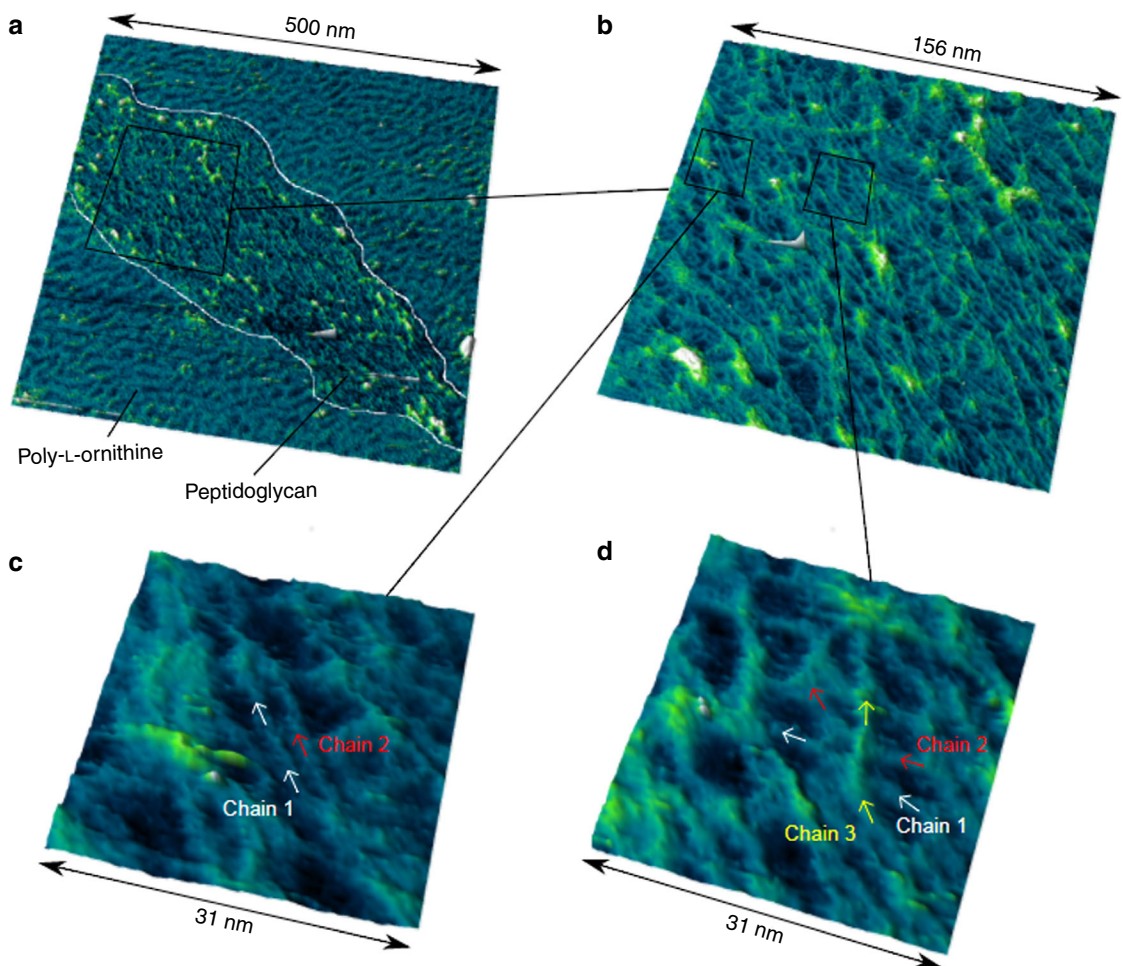

**Fig. 1** Direct visualisation of glycan strand arrangement in the *E. coli* polymer envelope by AFM. **a** 3D representation of a peptidoglycan fragment mounted on poly-L-ornithine (see annotations). **b** Higher resolution image of boxed region in **a**. Long glycan chains are visible. **c** Zoom of marked boxed region in **b** showing side-by-side chains. **d** Zoom of marked boxed region in **b** showing overlapping chains

generating an angular histogram of gradient orientations in our images and then fitting an ellipse to this (Fig. 2a, b). We defined the ratio of the long to the short axes of the ellipse as the orientational order parameter. An image with randomly oriented linear features is expected to have an orientational order parameter of ~1, which is the lowest possible value (see simulations, Supplementary Fig. 5). This approach gave a mean orientational order parameter for our experimental data of 1.6 (s. d. 0.1, $n = 6$). The method is simple and tolerant of AFM image noise and the nature of contrast at high resolution. Results were similar when an image was split into sub-regions and analysis carried out on each of these to yield a map of local orientational order (Supplementary Fig. 6) suggesting this parameter is fairly uniform—the mean value in the given example was 1.7 (s.d. 0.1, $n = 16$ sub-regions). It is the increased resolution in our current study (substantially improving on our previous work[7]) that has permitted us to determine this statistic. We used a segmentation-based approach to determine pore area distributions (Fig. 2c). This revealed pores up to 66 nm$^2$ in area (although most pores were less than 5 nm$^2$) supporting models of peptidoglycan biosynthesis requiring contact between inner and outer-membrane proteins through the polymer network[7,23,24].

**Glycan chain orientational order is pervasive.** Rod shape in *E. coli* is dependant on correctly functioning MreB multi-protein filaments[25], which move circumferentially around the internal surface of the inner cell membrane in a manner dependent on insertion of new peptidoglycan polymer[26]. MreB is linked to a proposed multi-enzyme nano-machine (elongasome), which spans the inner membrane such that catalytic enzymes can form bonds between new monomers and the pre-existing polymer. Much remains to be learned about how this process works—it is modulated by local surface curvature[27] and Lpo cofactors in the outer-membrane, which promote catalysis on contact with enzymes and may have a role in regulating peptidoglycan thickness[23,24]. We have previously proposed that regions of more porous peptidoglycan are more permissive for insertion of new material[7]. Cell poles are made during division in a process that does not involve MreB, but instead a similar trans-membrane system (divisome) involving the prokaryotic tubulin homologue FtsZ, which also moves circumferentially relative to the long axis of the cell[28,29]. In rod-shaped *E. coli* with a radius of 0.5 μm and a length of 2 μm, the percentage of the cylindrical surface area to the pole surface area, and thus an estimate of the amount of material contributed by the elongasome is 75% compared with 25% by the divisome, based on:

$$100\% = S_{cylinder} + S_{poles} = \frac{l}{\frac{4}{3}r + l} + \frac{\frac{4}{3}r}{\frac{4}{3}r + l}.$$ Thus, most of the

peptidoglycan is contributed by processes involving MreB, suggesting it might provide the link between peptidoglycan biosynthesis, architecture and cell morphology.

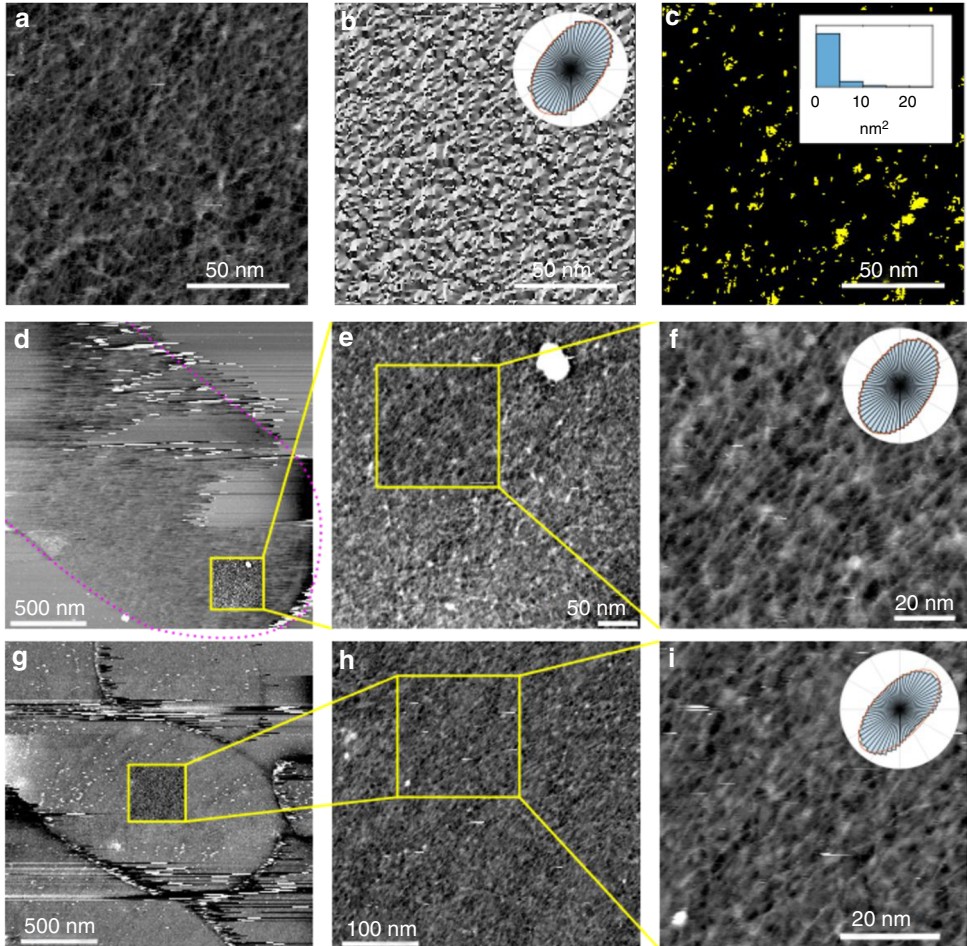

**Fig. 2** Mapping and quantification of glycan chain arrangement. **a** Example AFM image data. **b** Gradient orientation map of **a**. Inset: Polar histogram of data shown in **b**, overlaid with fitted ellipse. Dividing the long axis of the ellipse by the short yields a dimensionless parameter which reflects orientation order, with a higher number indicating an increased tendency to order. **c** Location of pores identified in the example image **a**. Inset: distribution of pore areas. **d** Peptidoglycan at pole: Large scan with bacterial shape outlined (dotted line). The instability inherent in imaging overlapping polymer leaflets is apparent in the top half of the image (height range 20 nm). **e** Smaller scan (acquired subsequently at lower scan speed) of region marked in **d** (height range 4 nm). **f** Enlargement of region marked in **e**. Many glycan strands running approximately along the circumferential axis of the cell are clearly visible (height range 3.5 nm). Inset: polar histogram. **g** Peptidoglycan from cylinder: Large scan of a different sacculus to **d** (height range 7.5 nm). **h** Smaller scan (acquired subsequently at lower scan speed) of region marked in **g** (height range 4 nm). **i** Enlargement of region marked in **h**. Many glycan strands running approximately along the circumferential axis of the cell are clearly visible (height range 4 nm). Inset: polar histogram

By developing a novel fixation protocol, we were able to stabilise the two leaflets of peptidoglycan in an intact polymer envelope sufficiently to image peptidoglycan from the cell poles (Fig. 2d–f; Supplementary Fig. 7a–f) and also the cylindrical part of the cell (Fig. 2g–i). This revealed that the poles had a similar chain organisation to the rest of the cell surface (without a statistically significant difference in orientational order (Supplementary Fig. 8a)) and apparent long glycan chains. This shows that peptidoglycan is structurally similar over the entire surface of the *E. coli* cell, even though the poles and the cylindrical part of the cell are biosynthesised by different sets of nano-machinery. It is striking that although in the cylindrical part of the cell glycan chains tend to be aligned with the direction of maximum stress, this is not the case in the hemispherical poles (where stress is directionally isotropic). Chain direction may be more a result of an evolutionary drive to efficiently create a rod shape than to create a material that resists anisotropic forces.

It is important to note that turgor-induced forces lead to stretching of peptidoglycan along the longitudinal axis of the cell in vivo[18,30], and perhaps circumferentially also[31,32]. However, we would not expect shear stresses, or other more complex stress

patterns to emerge. There is no substantial difference to overall chain orientation in a sacculus as compared to a living cell. There are, however, likely to be differences in chain spacing. Furthermore, organisation may be locally altered as strain varies across the network, e.g. a very porous region might enlarge more than a very dense region under turgor.

**E. coli peptidoglycan contains long glycan chains**. Final glycan chain length depends on the activity of glycosyltransferases that catalyse addition of new disaccharide monomers to the strands and peptidoglycan hydrolases that catalyse chain breaks. Chain length is also an important input into models and simulations of the cell wall.

To investigate the elongated features we observed in the extant peptidoglycan polymer envelope, we radiolabelled glycan strands with [$^{14}$C]-GlcNAc and digested purified peptidoglycan sacculi with ATL amidase[33] (which separates the glycans and peptides). Next, we used gel filtration chromatography to investigate the size distribution of glycan strands[5,21] (Fig. 3a). Overall, 66% of radioactive material eluted before a 100 kDa calibration standard

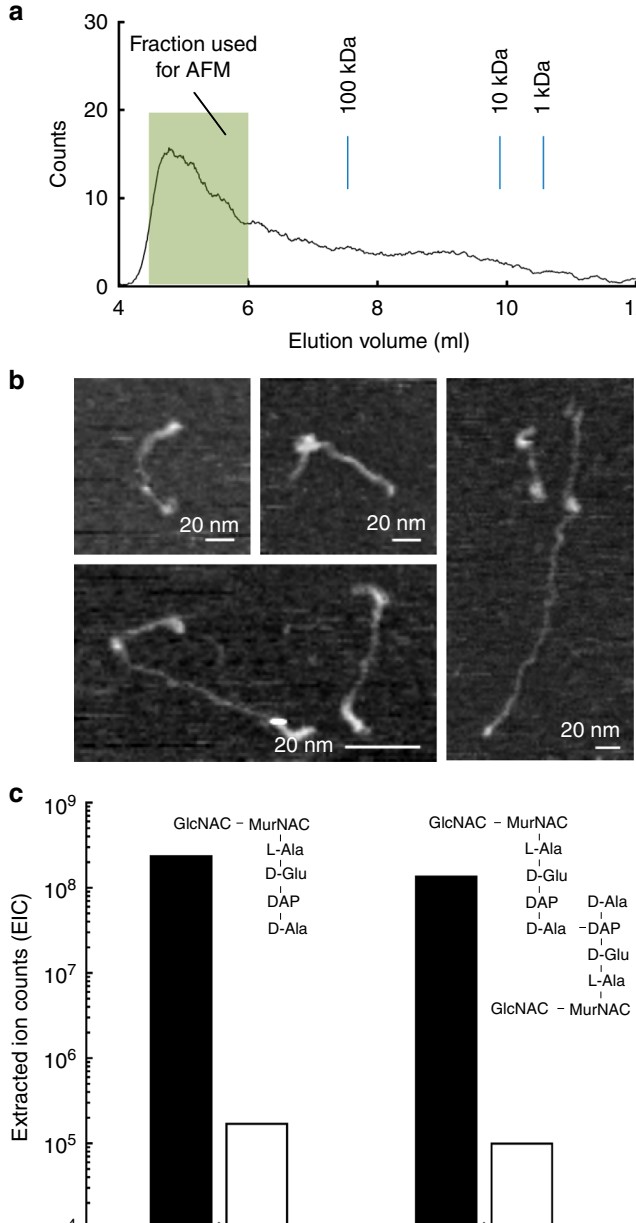

**Fig. 3** Size exclusion chromatography and AFM imaging of isolated glycan strands. **a** Size exclusion chromatography trace of glycan chains from *E. coli* (MG1655). **b** AFM images of glycan chains from fraction shown in **a** (each panel adjusted for best contrast). **c** Sums of extracted ion counts (EIC) for positively charged adducts of the major peptidoglycan sugar-peptide monomer and dimer from material digested with either Cellosyl alone (black bars), or ATL amidase, then Cellosyl (white bars). Amidase treatment removes the cross-linking peptides, substantially reducing the abundance of sugar-peptide adducts and freeing the glycan chains

(equivalent to a chain length of about 200 nm). Glycan chains from a high molecular weight fraction (Fig. 3a) were imaged directly by AFM (Fig. 3b) and were observed to be up to 200 nm in length, far higher than when measured previously by reverse-phase HPLC[22] which cannot directly measure high molecular weight material. This is the first observation of such long glycan chains in peptidoglycan from *E. coli* or any similar (Gram-negative) bacteria. A low estimate of the distances over which MreB moves (~200 nm[26]) are similar to the length of the longer glycan strands

we observed by AFM. We confirmed peptidoglycan digestion by ATL amidase using reversed-phase liquid chromatography coupled to mass spectrometry (RP-LC–MS). This showed that ATL amidase treatment led to a three orders of magnitude reduction in cross-links (Fig. 3c; Supplementary Fig. 2) indicating highly efficient removal of contaminating cross-linked material.

**Rod shape relates to long, orientationally ordered glycans.** Bacteria adopt a broad range of shapes[34] and a major assumption in the field is that different cell shapes require altered glycan chain organisation. To test this we generated *E. coli* cells that were approximately spherical by inhibiting MreB polymerisation with A22[35] or using a mutant lacking the gene encoding MreB[36] (Fig. 4a, b), and analysed peptidoglycan polymer characteristics. Cells lacking functional MreB did not have a "minicell" phenotype, i.e. they were not two poles (made by the FtsZ-mediated divisome machine) stuck together[37] as they had substantially larger diameters (>2 μm) than that of rod-shaped *E. coli* (~1 μm). The polymer networks from *E. coli* with altered shape had lost orientational order (Fig. 4c, d; Supplementary Fig. 8): Evidence that the micron-scale morphological features of *E. coli* depend on the nanoscale organisation of the peptidoglycan polymer envelope. Lack of functional MreB also led to a concomitant reduction in glycan strand length with=32% of radioactive material eluted before a 100 kDa calibration standard for bacteria grown in media containing 10 μg/ml A22 (Fig. 4e), 36% for the strain lacking the gene encoding MreB (Fig. 4f), compared with 66% for wild-type/untreated. Interestingly, A22 has previously been found to reduce glycan chain length in a different species of bacteria: *Caulobacter crescentus*[38], without resolving very high molecular weight material. With mutated *mreC*, *Bacillus subtilis* has much shorter glycan chains[5]. We propose that disruption of systems directing peptidoglycan synthesis will generally result in reduced glycan chain length. As we have shown here that glycan chain lengths can vary greatly without leading to the death of cells, the same is not true for pore sizes which were similar for "rod" and "sphere" shaped *E. coli* (Supplementary Fig. 9). Excessively large pores would likely cause major problems for the integrity of the polymer envelope.

Without functional MreB, *E. coli* cells are still able to biosynthesize peptidoglycan permitting cellular enlargement. This may involve components of the elongasome and/or divisome working independently of MreB. We propose that this less well oriented monomer addition involves lower processivity of enzymatic catalysis of glycosidic bonds, leading to shorter glycan chains. This makes the polymer more mechanically isotropic and less able to maintain a rod shape with associated loss of polarity and chains being inserted at a greater range of angles. It has recently been discovered that in *B. subtilis* MreB orients along the direction of greatest membrane curvature[39]—locally establishing and maintaining rod shape (via the elongasome). This provides insight into the underlying molecular mechanism by which peptidoglycan with varying degrees of orientational order can be synthesised.

**Discussion**
As we have shown here, direct visualisation of chain arrangement in a complex biopolymer network provides new insights into how such materials perform essential functions within a cell. Rod-shaped *E. coli* is characterised by long glycan chains and more ordered peptidoglycan, whereas spheroid shaped *E. coli* has shorter chains that are significantly less ordered (Fig. 5). In future, we envisage our new methodology can be applied to investigate polymer network organisation in a large number of important biological samples (including different species of bacteria, plants

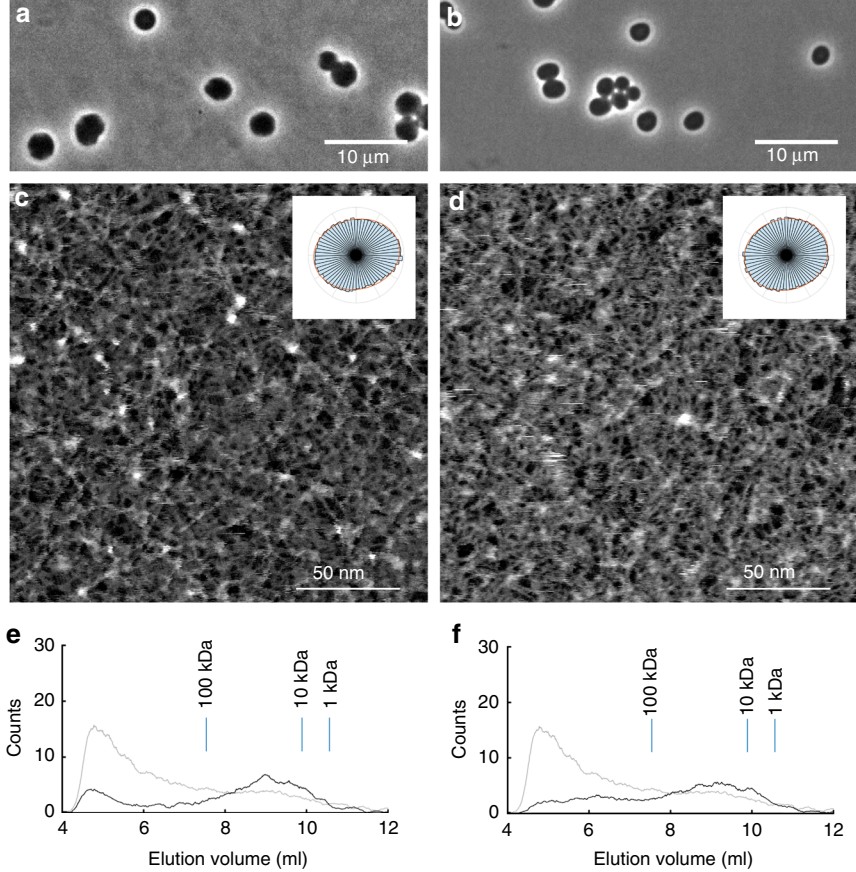

**Fig. 4** Peptidoglycan characteristics in spheroid *E. coli*. **a** Optical phase contrast microscopy image of *E. coli* bacterial cells grown in media containing 10 μg/ml A22. **b** Optical phase contrast microscopy image of *E. coli* bacterial cells lacking the gene encoding MreB. **c** Image of peptidoglycan from *E. coli* grown in media containing A22, and associated polar histogram. Glycan strands appear less orientationally ordered and this is reflected in the more circular shape of the polar histogram. **d** Image of peptidoglycan from *E. coli* lacking the gene encoding MreB, and associated polar histogram. Glycan strands again appear less orientationally ordered. **e** Size exclusion chromatography trace of glycan chains from *E. coli* grown in media containing 10 μg/ml A22 (black line). Chains from bacteria grown in this way are shorter than for bacteria grown in unmodified media (see trace in grey for comparison). **f** Size exclusion chromatography trace of glycan chains from *E. coli* lacking the gene encoding MreB (black line). Chains from this genetically modified bacterial strain are shorter than those of unmodified bacteria (grey line)

and fungi) and bio-inspired nanostructures, and connect this to material function.

## Methods

**Growth of bacteria**. Bacteria (*E. coli* MG1655 [kindly provided by Jeff Green] or MC1000 Δ*mreB* [kindly provided by Kenn Gerdes]) were grown in lysogeny broth (supplemented with 10 μg/ml A22 where stated) at 37 °C with agitation at 200 rpm.

**Purification of peptidoglycan polymer envelopes**. Peptidoglycan "sacculi" were purified using established methods[40]. Briefly, cells were grown to exponential phase and killed by boiling in SDS solution. The SDS was then washed out by ultra-centrifugation before treatment with trypsin (protease), further boiling in SDS and washing.

The SDS treatment frees peptidoglycan from many of the other constituents of the bacterial cell. Peptidoglycan is not a protein, so this does not cause refolding problems. Trypsin does not affect peptidoglycan, but digests any protein bound to it that might otherwise obscure the material of interest.

For [14]C-labelled peptidoglycan, *N*-acetyl[14C]glucosamine ([14C]GlcNAc) was added to the culture medium, as previously described[21].

**Mounting of polymer envelopes for AFM imaging**. Mica discs were mounted on steel stubs and cleaved to reveal a clean, flat substrate. Then a drop of poly-L-ornithine (3 μg/ml in water) was pipetted onto this, before the surface was washed three times with water.

Poly-L-ornithine (and poly-L-lysine) have been used extensively for immobilising cells and biomolecules (e.g. DNA) for high resolution imaging. The forces between poly-L-ornithine and peptidoglycan are highly unlikely to be strong

enough to disrupt the organisation of the peptidoglycan itself, which is mainly determined by covalent bonds between peptides and sugars.

After poly-L-ornithine treatment, a drop of peptidoglycan suspension at a previously determined working concentration (determined empirically for each batch) in 10 mM Tris (pH 8) was added to the substrate and incubated for 3 minutes, without allowing it to dry. This was then washed three times with 10 mM Tris (pH 8).

To image pole/cylinder peptidoglycan, the samples were subsequently incubated in 2.5% (w:v) glutaraldehyde in water (10 min) and then washed a further three times before imaging.

Glutaraldehyde cross-links amine groups which are present in the diaminopimelic acids of the peptidoglycan cross-link peptides and are abundant in the poly-L-ornithine. The result is that the peptidoglycan becomes more tightly bound to the substrate and itself, allowing double leaflets to be imaged.

The samples were not allowed to dry out at any time.

**AFM imaging of polymer envelopes**. Imaging was carried out using a Bruker Dimension FastScan AFM using FastScan-D probes (Bruker—nominal $k = 0.25$ N/m, nominal cantilever length = 16 μm) in "Tapping Mode" (Amplitude Modulated Intermittent Contact Mode) driven at ~110 kHz with a free amplitude of ~1 nm.

All imaging of polymer envelopes (sacculi) was carried out in liquid (buffer) without drying at any point.

**Purification of glycan strands**. Peptidoglycan was digested using recombinant ATL amidase or Cellosyl, as previously described[21]. Briefly, [14]C-labelled pepti-doglycan was purified as described above. It was then reacted overnight at 37 °C with at least 10× the amount of enzyme previously determined to be required to completely digest the substrate. The resulting mixture was then boiled for 10 min to

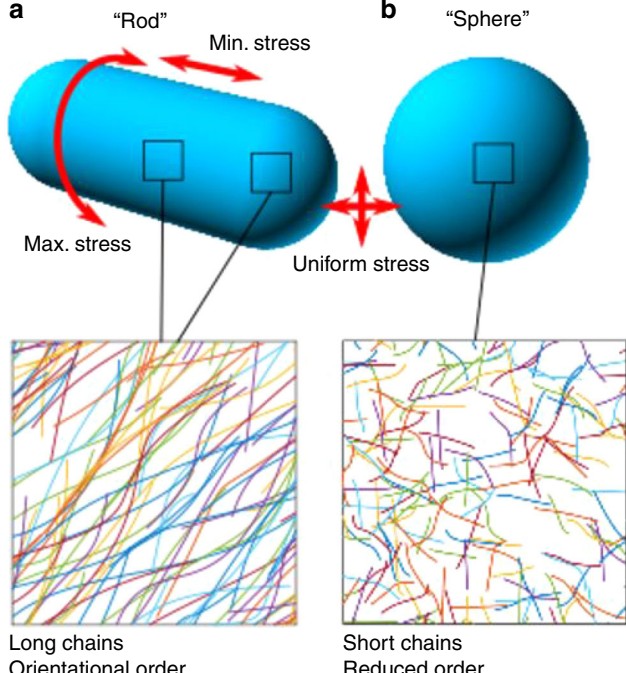

**Fig. 5** Conceptual diagrams. **a** Peptidoglycan from rod-shaped *E. coli* (MG1655) is not crystalline yet has quantifiable orientational order with chains likely to be in a circumferential direction. This corresponds to the direction of maximum stress in the cylindrical part of the cell (stress is isotropic at the poles). It contains glycans up to 200 nm long. **b** Peptidoglycan from roughly spherical *E. coli* (lacking MreB or treated with A22) is much less ordered and has shorter glycan chains. Stress is isotropic in this case

Images were "line flattened" and a plane fitted and subtracted using Gwyddion (http://gwyddion.net). Data was loaded into Matlab and downsampled 0.5-fold such that four pixels in the parent image were averaged to a single pixel (this further reduced the influence of scan line noise on the subsequent analysis). This image was then numerically differentiated with respect to $x$ and $y$ yielding two matrices from which a gradient vector could be determined at each pixel. A polar histogram of gradient orientations was then plotted and an ellipse fitted to it. We defined the ratio of the long to the short axis of the ellipse as the orientational order parameter.

**Simulation of test images**. Linear features were generated with random start points, fixed lengths and a distribution of angular orientations of a defined standard deviation. This image was convolved with a Gaussian kernel to simulate tip convolution. To the result we added "scan line noise" where each horizontal line was given a random offset drawn from a normal distribution, and Gaussian noise. The intention was not to physically simulate the AFM and the unique interaction between peptidoglycan and the AFM probe, but to validate our image analysis methods.

**Determination of pore statistics**. An adaptive threshold was applied to images to identify pores (Matlab implementation of Bradley's method[41]). The resulting binary images were then morphologically filled and then opened to remove regions of four or less pixels (to set a minimum definition of pore size).

**Optical microscopy**. Bacteria were fixed with paraformaldehyde, as previously described[7], then mounted on an agarose pad before imaging with a Nikon Ti Eclipse inverted optical microscope in phase contrast mode.

**Code availability**. The code used in this study is available from the corresponding author on request.

**Data availability**. The data supporting the findings of the study are available in this article and its supplementary information files, or from the corresponding author on request.

deactivate the enzyme, before centrifugation to remove denatured enzyme. Samples were stored at −20 °C until required, if not used immediately.

**Size exclusion chromatography**. This was carried out as previously described[21]. Approximately 50,000 cpm of radiolabelled peptidoglycan was injected in a volume of 50 µl onto a TSKgel G 4000 SW column ($l = 300$, $d = 7.5$ mm) which had previously been equilibrated in sodium phosphate buffer (100 mM, pH 6). Flow rate was 0.3 ml/min. Radioactivity was detected using a β-RAM (LabLogic) flow scintillation counter equipped with a 100 µl cell and running a 1:1 sample:scintillation cocktail mix. Calibration was carried out using dextran standards.

To extract strands for AFM imaging, unlabelled material was injected and the eluent collected rather than being directed to the β-RAM. The elution buffer in this case was 10 mM sodium phosphate (pH 2).

**AFM imaging of glycan strands**. Material collected by size exclusion chromatography was diluted to an appropriate working concentration (determined empirically for each batch) in water. A drop was applied to mica and then immediately dried with flowing nitrogen.

Imaging of glycan strands was carried out in air using a Bruker Dimension FastScan AFM with TESPA-V2 probes (Bruker—nominal $k = 37$ N/m, nominal cantilever length = 123 µm) in "Tapping Mode" (Amplitude Modulated Intermittent Contact Mode) driven at ~320 kHz. No other AFM in this study was carried out in air.

**Peptidoglycan structural analysis by LC–MS**. Sacculi (or glycan chains) were digested with cellosyl, boiled to deactivate enzyme and then reduced with sodium borohydride as previously described[5]. The resulting soluble material was separated by reverse-phase high-performance liquid chromatography (RP-HPLC) using a Hypersil aQ C18 column (3 µm, 2.1 by 200 mm) coupled to an Agilent 6500 series quadrupole time of flight mass spectrometer (Q-TOF LC–MS). Flow rate was 0.15 ml/min. Buffer A was water and buffer B was acetonitrile, both containing 0.1% (v:v) formic acid. The gradient was 0 to 15% buffer B over 35 min.

**Gradient orientation analysis of orientational order**. This was achieved using a combination of the Gwyddion open source software and custom Matlab scripts.

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

## Acknowledgements

We thank Sandip Kumar, Nicolas Olivier and Nic Mullin for useful discussions, Chris Thoroughgood for providing the ATL amidase enzyme and Simon Thorpe for RP-LC–MS analysis. This work was supported by the BBSRC (BB/L006162/1, BB/N000951/1, BB/L014904/1) and MRC (MR/N002679/1). Optical imaging was done in the Wolfson Light Microscopy Facility and the super-resolution facility in Sheffield was funded by MRC grant MR/K015753/1.

## Author contributions

R.D.T., S.M., J.K.H. and S.J.F. contributed to study design, data analysis and manuscript preparation. R.D.T. and S.M. also carried out experimental work.

## Additional information

**Competing interests:** The authors declare no competing interests.

