## [Peer Review File(PDF 180 kb) · Nature Communications]

Reviewers' comments:

Reviewer #1 (Remarks to the Author):

This manuscript by Turner et al describes advances in AFM imaging of peptidoglycan of E. coli cells, as well as some biochemical characterization of the peptidoglycan. The primary conclusions are that the cell wall contains glycan strands longer than previously detected, that glycan strands near the poles are circumferentially aligned, as expected for the cylindrical portion of the wall, and that conditions resulting in a change to disordered (spherical) morphology is tied to the generation of disordered peptidoglycan.

Assuming that the authors interpretations of the images are correct, then the methods and potential for future experimentation are important for the field. The method for quantification of strand orientation preference within such a complex image appears logical and useful.

My concern is that there appears to be a few gaps or oversights within the presentation or logic. Some of these might be easily fixed and would strengthen and clarify the analyses for the general audience.

1) The paper needs some convincing evidence or argument that what is being visualized within the walls are glycan strands. The orientation of the strands within the polar region is helpful in this regard. If this could be done in at least one case for a region within the cylindrical part of the wall, it would tie everything together. The organizational arrangement observed in the PG fragments tells us nothing about whether that alignment is circumferential versus longitudinal. If it could be shown to be circumferential, as suggested by previous publications, then the entire argument would be tied together. For a cell such as that visualized in Fig 2d, can't a region of that cell within the cylindrical portion of the cell be examined in the same highly detailed fashion? Even in one case where such a sacculus might be stably mounted and crosslinked?

2) For figure 1 it is stated that sacculus fragments are being observed. What is the evidence or logical argument that we are looking at fragments versus sacculi?

3) Figure 3b: What is the evidence that we are visualizing glycan strands following size exclusion chromatography? Were these strands analyzed in any other way? Is there any analysis to demonstrate a lack of any contaminating material?

4) Page 7, lines 5-6: To place the 32% and 36% in context, it would be helpful to restate the 66% observed in wild type cells or the % found within the controls for this particular experiment.

5) Figure 5 is never cited or described in the text.

6) Page 16: In the description of AFM imaging, it is not stated if this was done wet or "in air" as for the free glycan strands.

Reviewer #2 (Remarks to the Author):

The molecular organization of peptidoglycan in the bacterial cell wall has been the subject of intense interest for decades. However, despite many biochemical, molecular and computational approaches, the true three-dimensional architecture of this fundamental structure is still unclear. In this work, the authors use atomic force microscopy to image individual glycan strands directly. These strands are not organized in a perfectly regular geometry (as is often depicted in textbooks and models), but the strands do have a preferred orientation that wraps around the cell girth (matching the general view of

most models). In addition, the authors visualized previously unseen, very long glycan strands in cell walls and in fractions of purified peptidoglycan. The work is an important technical advance and well presented, and validates an approach that should be valuable for examining the cell walls of many organisms across all biological kingdoms.

Specific comments

1. Although the work is definitely a leap forward in understanding the in vivo organization of the cell wall, the authors should discuss the fact that these images were made on isolated cell walls that were not under turgor pressure. Thus, it remains possible that the strand organization might be oriented differently in a living cell.

2. The existence of extremely long glycan chains is interesting. These were observed but not quantified in previous HPLC studies. Can the authors estimate the percentage or fraction of peptidoglycan that is in these long strands, and compare that with these prior results?

Minor comments

3. The authors state on page 3 that “some chains overlap each other (Fig. 1f)...” Did the authors mean to say Fig. 1e or 1h? The legend in Figure 1 states that panel f consists of “Two glycan chains side by side...”

4. The following phrasing at the bottom of page 3 is awkward: “After investigating several approaches to quantification of the network”

5. Figure 1E. Is there any way to produce line profiles from the image? This would greatly aid the reader, since the image is understandably fuzzy and difficult for the neophyte to interpret.

6. Figure 1E: Could the authors comment on the denser regions (i.e. white splotches) that appear at the confluence of several glycan strands? These bright white patches also appear in images of isolated glycan strands in Fig. 3B.

7. Figure 1H: It is not readily apparent that the strands are overlapping. Would line profiles help to make the point?

8. Supplementary Figure 6. In panel “a”, please change the x-axis typography so that “wild type” and “A22” don’t appear to run together as though they were one phrase (which is confusing). Also, please note that the “Poles” are themselves also from wild type sacculi. Perhaps it would be clearer if the two “wild type” samples were placed together on the left side. This way they could be differentiated as being from the “cylinder” and the “poles” of wild type sacculi.

Reviewer #3 (Remarks to the Author):

The authors published many nice AFM images recorded with the skills of experienced experimentalists in this study, but unfortunately, it is unclear to this reviewer how much structure and arrangement of the glycan chains could reflect the 'in vivo' situation, as purification and imaging conditions appear to be pretty harsh:

Peptidoglycans were purified in SDS. How can we be sure that refolding occurred correctly when

removing the detergent? What happened during trypsin digestion? What during fixation with gluteraldehyde? How does the surface bound poly-L-ornithine influence surface arrangement. The authors state that imaging was performed in the air, but the sample was not allowed to dry at any point. How does this fit together?

With these conditions, how sure can the authors be that the 'textbook crystalline model' is wrong?

Reviewers' comments:

Reviewer #1 (Remarks to the Author):

This manuscript by Turner et al describes advances in AFM imaging of peptidoglycan of *E. coli* cells, as well as some biochemical characterization of the peptidoglycan. The primary conclusions are that the cell wall contains glycan strands longer than previously detected, that glycan strands near the poles are circumferentially aligned, as expected for the cylindrical portion of the wall, and that conditions resulting in a change to disordered (spherical) morphology is tied to the generation of disordered peptidoglycan.

Assuming that the authors' interpretations of the images are correct, then the methods and potential for future experimentation are important for the field. The method for quantification of strand orientation preference within such a complex image appears logical and useful.

My concern is that there appears to be a few gaps or oversights within the presentation or logic. Some of these might be easily fixed and would strengthen and clarify the analyses for the general audience.

1) The paper needs some convincing evidence or argument that what is being visualized within the walls are glycan strands. The orientation of the strands within the polar region is helpful in this regard. If this could be done in at least one case for a region within the cylindrical part of the wall, it would tie everything together. The organizational arrangement observed in the PG fragments tells us nothing about whether that alignment is circumferential versus longitudinal. If it could be shown to be circumferential, as suggested by previous publications, then the entire argument would be tied together. For a cell such as that visualized in Fig 2d, can't a region of that cell within the cylindrical portion of the cell be examined in the same highly detailed fashion? Even in one case where such a sacculus might be stably mounted and crosslinked?

We appreciate this constructive comment and have now imaged regions from the cylindrical portion of the cell and find a similar architecture to poles and fragments (Fig 2 and Fig S8a).

2) For figure 1 it is stated that sacculus fragments are being observed. What is the evidence or logical argument that we are looking at fragments versus sacculi?

We have revised figure 1 to show an enlarged image of a sacculus fragment, and made our explanation clearer (lines 53-55). Fragmentation of *E. coli* sacculi for microscopy has previously been carried out^{1,2} and fragments are clearly distinguishable from intact sacculi due to their morphology.

3) Figure 3b: What is the evidence that we are visualizing glycan strands following size exclusion chromatography? Were these strands analyzed in any other way? Is there any analysis to demonstrate a lack of any contaminating material?

We use established methods to obtain pure peptidoglycan, which after digestion with ATL amidase produces a mixture of enzyme, peptides and glycans. We take only the highest molecular weight material for our AFM analysis. ATL amidase (~60 kDa) and the cross-link peptides are eluted later in the trace. Most importantly, digestion of glycan with a muramidase (Cellosyl) generates low molecular weight (late eluting) products. This shows that the high molecular weight radiolabeled material was glycan (TSK2000 size exclusion chromatography traces below and Fig S4).

4) Page 7, lines 5-6: To place the 32% and 36% in context, it would be helpful to restate the 66% observed in wild type cells or the % found within the controls for this particular experiment.

We have amended the text to include 66% in context (line 178).

5) Figure 5 is never cited or described in the text.

We have added a citation in final paragraph of manuscript (line 200).

6) Page 16: In the description of AFM imaging, it is not stated if this was done wet or “in air” as for the free glycan strands.

We have made it clear in the method section (line 294) and throughout that AFM imaging of sacculi was all done in buffer.

Reviewer #2 (Remarks to the Author):

The molecular organization of peptidoglycan in the bacterial cell wall has been the subject of intense interest for decades. However, despite many biochemical, molecular and computational approaches, the true three-dimensional architecture of this fundamental structure is still unclear. In this work, the authors use atomic force microscopy to image individual glycan strands directly. These strands are not organized in a perfectly regular geometry (as is often depicted in textbooks and models), but the strands do have a preferred orientation that wraps around the cell girth (matching the general view of most models). In addition, the authors visualized previously unseen, very long glycan strands in cell walls and in fractions of purified peptidoglycan. The work is an important technical advance and well presented, and validates an approach that should be valuable for examining the cell walls of many organisms across all biological kingdoms.

Specific comments

1. Although the work is definitely a leap forward in understanding the in vivo organization of the cell wall, the authors should discuss the fact that these images were made on isolated cell walls that were not under turgor pressure. Thus, it remains possible that the strand organization might be oriented differently in a living cell.

Peptidoglycan experiences strain due to turgor pressure in a living cell and this must be a consideration when interpreting our data. We expect that in a living cell, the network we see in a sacculus would be stretched.

There is no consensus the amount of strain and degree of difference between longitudinal and circumferential peptidoglycan strain in *E. coli*. Although it is accepted that longitudinal strain is greater than circumferential strain. A

figure of 17%³ for longitudinal strain is well cited. Electron cryotomography results show there is no circumferential strain, at least in *Caulobacter crescentus*⁴. More recent optical microscopy studies suggest some circumferential strain may be present^{5,6}.

The *in vivo* stress distribution is anisotropic, with circumferential stress twice longitudinal stress. The network would therefore be stretched both perpendicular to the general direction of the glycan chains and possibly also along it. However, we do not expect shear stresses, or other more complex stress patterns to emerge (if we take *E. coli* to be a cylindrical pressure vessel with hemispherical caps). There will therefore be no substantial difference to overall chain orientation in a sacculus as compared to a living cell. There may, however, be differences in organisation as total strain varies across the network e.g. a very porous region might enlarge more than a very dense region under turgor.

We have added a discussion of this to our manuscript (lines 139-145).

2. The existence of extremely long glycan chains is interesting. These were observed but not quantified in previous HPLC studies. Can the authors estimate the percentage or fraction of peptidoglycan that is in these long strands, and compare that with these prior results?

We found 66% of radioactive material eluted before a 100 kDa dextran standard (which equates to chains over 200 nm long). Previously no upper bound had been placed upon chain length - a key aspect of our findings is the confirmation by AFM that very long chain are present. Direct quantitative comparisons between our combined size exclusion chromatography and AFM approach cannot be made with previous HPLC studies due to methodological differences. Previous quantitative studies have shown significant amounts of unresolved high molecular weight material⁷ (lines 155-158).

Minor comments

3. The authors state on page 3 that “some chains overlap each other (Fig. 1f)...” Did the authors mean to say Fig. 1e or 1h? The legend in Figure 1 states that panel f consists of “Two glycan chains side by side...”

The reviewer is right. We have corrected manuscript text (line 77), although this figure is now in supplementary information.

4. The following phrasing at the bottom of page 3 is awkward: “After investigating several approaches to quantification of the network”

We have amended the text to address this (line 85).

5. Figure 1E. Is there any way to produce line profiles from the image? This would greatly aid the reader, since the image is understandably fuzzy and difficult for the neophyte to interpret.

We have changed the way in which this data is presented and hope it is now clearer (Fig 1).

6. Figure 1E: Could the authors comment on the denser regions (i.e. white splotches) that appear at the confluence of several glycan strands? These bright white patches also appear in images of isolated glycan strands in Fig. 3B.

White splotches are likely regions where strands overlap and are too close together or insufficiently well immobilised to resolve. They are topographically higher up than darker regions. The change to the way in which some of the data is presented facilitates interpretation (Fig 1).

7. Figure 1H: It is not readily apparent that the strands are overlapping. Would line profiles help to make the point?

The new way of presenting AFM data in the revised manuscript makes this clearer (Fig 1c).

8. Supplementary Figure 6. In panel “a”, please change the x-axis typography so that “wild type” and “A22” don’t appear to run together as though they were one phrase (which is confusing). Also, please note that the “Poles” are

themselves also from wild type sacculi. Perhaps it would be clearer if the two “wild type” samples were placed together on the left side. This way they could be differentiated as being from the “cylinder” and the “poles” of wild type sacculi.

Now having data for the ‘cylindrical’ part of intact sacculi using our post-fixing method (see Reviewer 1, Comment 1), we are able to make separate comparisons between fragments of wild type, wild type treated with A22 and the *mreB* delete strain, and between pole and cylinder peptidoglycan (Figure S8). This confirms our earlier results and is a clearer presentation of the data.

Reviewer #3 (Remarks to the Author):

The authors published many nice AFM images recorded with the skills of experienced experimentalists in this study, but unfortunately, it is unclear to this reviewer how much structure and arrangement of the glycan chains could reflect the 'in vivo' situation, as purification and imaging conditions appear to be pretty harsh:

Peptidoglycans were purified in SDS. How can we be sure that refolding occurred correctly when removing the detergent?

Peptidoglycan structure is primarily dictated by covalent bonds. There is no indication that it folds like a protein, so refolding is not an issue.

What happened during trypsin digestion?

Trypsin is a protease that will digest proteins ionically and covalently bound to the peptidoglycan. This is an extremely important step as protein contamination of the material would otherwise obscure our images. The peptidoglycan itself is unaffected by trypsin.

What during fixation with glutaraldehyde?

Glutaraldehyde fixation cross-links amine groups. These appear in the diaminopimelic acids of the cross-linking peptides, and are abundant in the poly-L-ornithine. This process is unlikely to be 100% efficient, so there will still be some free amine groups after treatment. The result is that the peptidoglycan becomes more tightly bound to the substrate and itself, allowing double leaflets to be imaged.

How does the surface bound poly-L-ornithine influence surface arrangement?

The structure of peptidoglycan is primarily due to covalent bonding. The attractive forces between the poly-L-ornithine and the peptidoglycan would not be sufficiently high to break these bonds and substantially alter structure. Poly-L-ornithine (or poly-L-lysine) are routinely used as a means of adhering biological material to surfaces for imaging, including optical microscopy of cells and AFM imaging of DNA. Our current data is in good agreement with previous AFM images obtained in air or liquid without any prior surface treatment (i.e. no poly-L-ornithine)², albeit at much better resolution.

The authors state that imaging was performed in the air, but the sample was not allowed to dry at any point. How does this fit together?

We have clarified the text throughout the manuscript. All imaging was done in liquid (buffer), apart from the isolated glycan strands.

With these conditions, how sure can the authors be that the 'textbook crystalline model' is wrong?

We are confident that our findings supersede the textbook model. The purification process (which would be harsh for most proteins, and irreversibly destructive to membranes) is completely standard and represents the current state of the art for obtaining pure peptidoglycan. We expect that isolated peptidoglycan will have shrunk somewhat compared to in vivo material (see response to reviewer 2, comment 1, above), and we concede that any hypothetical secondary structure (e.g. hydrogen bonding) will have been disrupted. However, the principal

structure, critical to our study and dictated by covalent bonds, is unperturbed by our methods. Furthermore, we can not only quantify differences in peptidoglycan from untreated / wild type bacteria and A22 treated / MreB-deleted strain, but we can explain these plausibly in the context of bacterial cell biology. DNA is purified in solvents that denature proteins, but has nonetheless also been extensively imaged by AFM in buffer at very high resolutions (a good example in this reference ⁸).

However, we agree that the treatment of peptidoglycan in our study is apparently harsh and we did not explain this fully. We have now made modified the text to explain the purification and immobilisation process and its effects (lines 48-52, 139-145, 240-245, 271-274, 280-283, 290-293). We have also added caveats to indicate where our data is limited and that it must be understood in the context of the way in which it has been obtained.

References

1. Verwer, R. W., Beachey, E. H., Keck, W., Stoub, A. M. & Poldermans, J. E. Oriented fragmentation of *Escherichia coli* sacculi by sonication. *J. Bacteriol.* **141**, 327–332 (1980).
2. Turner, R. D., Hurd, A. F., Cadby, A., Hobbs, J. K. & Foster, S. J. Cell wall elongation mode in Gram-negative bacteria is determined by peptidoglycan architecture. *Nat. Commun.* **4**, 1496 (2013).
3. Koch, A. L., Lane, S. L., Miller, J. A. & Nickens, D. G. Contraction of filaments of *Escherichia coli* after disruption of cell membrane by detergent. *J. Bacteriol.* **169**, 1979–1984 (1987).
4. Gan, L., Chen, S. & Jensen, G. J. Molecular organization of Gram-negative peptidoglycan. *Proc. Natl. Acad. Sci. U. S. A.* **105**, 18953–18957 (2008).
5. Pilizota, T. & Shaevitz, J. W. Fast, multiphase volume adaptation to hyperosmotic shock by *Escherichia coli*. *PLoS One* **7**, e35205 (2012).
6. Rojas, E., Theriot, J. A. & Huang, K. C. Response of *Escherichia coli* growth rate to osmotic shock. *Proc. Natl. Acad. Sci. U. S. A.* **111**, 7807–7812 (2014).
7. Harz, H., Burgdorf, K. & Höltje, J. V. Isolation and separation of the glycan strands from murein of *Escherichia coli* by reversed-phase high-performance liquid chromatography. *Anal. Biochem.* **190**, 120–128 (1990).
8. Leung, C. *et al.* Atomic force microscopy with nanoscale cantilevers resolves different structural conformations of the DNA double helix. *Nano Lett.* **12**, 3846–3850 (2012).

REVIEWERS' COMMENTS:

Reviewer #1 (Remarks to the Author):

The authors have nicely addressed the reviewers concerns and have added explanatory comments to the manuscript that should allow th greater appreciation by a wider audience.

A minor comment:

Line 152: should say "used gel filtration chromatography 'to' investigate"

Reviewer #2 (Remarks to the Author):

The authors have edited the manuscript appropriately in response to my earlier comments. I have no other comments, and feel the manuscript is ready.

Reviewer #3 (Remarks to the Author):

The authors have answered all my questions on a scientifically fair basis and argued with their best effort that their purification and modification steps still allow conclusions to be drawn for the native system.

Please find our point-by-point response below:

REVIEWERS' COMMENTS:

Reviewer #1 (Remarks to the Author):

The authors have nicely addressed the reviewers concerns and have added explanatory comments to the manuscript that should allow th greater appreciation by a wider audience.

A minor comment:

Line 152: should say "used gel filtration chromatography 'to' investigate"

We have added the word "to".

Reviewer #2 (Remarks to the Author):

The authors have edited the manuscript appropriately in response to my earlier comments. I have no other comments, and feel the manuscript is ready.

Reviewer #3 (Remarks to the Author):

The authors have answered all my questions on a scientifically fair basis and argued with their best effort that their purification and modification steps still allow conclusions to be drawn for the native system.